# Coseismic Deformation Field and Fault Slip Distribution Inversion of the 2020 Jiashi *Ms* 6.4 Earthquake: Considering the Atmospheric Effect with Sentinel-1 Data Interferometry

**DOI:** 10.3390/s23063046

**Published:** 2023-03-11

**Authors:** Xuedong Zhang, Jiaojie Li, Xianglei Liu, Ziqi Li, Nilufar Adil

**Affiliations:** 1School of Geomatics and Urban Spatial Informatics, Beijing University of Civil Engineering and Architecture, Beijing 100044, China; 2Key Laboratory for Urban Geomatics of National Administration of Surveying, Mapping and Geoinformation, Beijing 100044, China; 3BGI Engineering Consultants Ltd., Beijing 100038, China

**Keywords:** Jiashi earthquake, atmospheric effect, DInSAR, coseismic deformation, slip distribution, inversion

## Abstract

Due to some limitations associated with the atmospheric residual phase in Sentinel-1 data interferometry during the Jiashi earthquake, the detailed spatial distribution of the line-of-sight (LOS) surface deformation field is still not fully understood. This study, therefore, proposes an inversion method of coseismic deformation field and fault slip distribution, taking atmospheric effect into account to address this issue. First, an improved inverse distance weighted (IDW) interpolation tropospheric decomposition model is utilised to accurately estimate the turbulence component in tropospheric delay. Using the joint constraints of the corrected deformation fields, the geometric parameters of the seismogenic fault and the distribution of coseismic slip are then inverted. The findings show that the coseismic deformation field (long axis strike was nearly east–west) was distributed along the Kalpingtag fault and the Ozgertaou fault, and the earthquake was found to occur in the low dip thrust nappe structural belt at the subduction interface of the block. Correspondingly, the slip model further revealed that the slips were concentrated at depths between 10 and 20 km, with a maximum slip of 0.34 m. Accordingly, the seismic magnitude of the earthquake was estimated to be *Ms* 6.06. Considering the geological structure in the earthquake region and the fault source parameters, we infer that the Kepingtag reverse fault is responsible for the earthquake, and the improved IDW interpolation tropospheric decomposition model can perform atmospheric correction more effectively, which is also beneficial for the source parameter inversion of the Jiashi earthquake.

## 1. Introduction

Western China’s Jiashi County experienced a 6.4 *Ms* earthquake on 19 January 2020, which resulted in a number of casualties (one fatality and two minor injuries) and building damage (some houses had wall cracks and collapsed walls). Although the earthquake was on the northwestern edge of Tarim Basin at the intersection of the South Tianshan fold belt, Parmir arcuate tectonics, and Tarim Basin, there are numerous active Cenozoic faults with varying properties in this region, in addition to mountains and basins that coexist [1,2,3]. Since the tropospheric atmospheric delay will reduce the accuracy of the obtained seismic coseismic deformation field because of the strong topographic fluctuations in this area, it is necessary to take steps to mitigate the impact of atmospheric delay [4]. However, the majority of existing studies rely on Differential Interferometric Synthetic Aperture Radar (DInSAR) technology to directly obtain the seismic coseismic deformation field and frequently disregard the effect of the tropospheric delay phase on the final result [3,5,6]. In addition, the preliminary correction of the tropospheric delay phase in the Jiashi seismic zone revealed that the fitting residual of deformation data obtained by near-field regional simulation is approximately 2 cm [7]. Therefore, it is necessary to correct the tropospheric delay of the interferometric deformation field to reduce the influence of the atmospheric effect and improve the accuracy of the geometric parameters of the seismogenic fault and the inversion of the coseismic slip distribution prior to employing DInSAR technology to address the structure and special terrain of the Jiashi earthquake.

Existing research classifies tropospheric atmospheric delay correction techniques into two categories [8,9,10,11,12]. One is to employ SAR data for filtering corrections. This method primarily accomplishes tropospheric delay phase estimation and removal from interferograms via filtering, correlation analysis, or averaging concepts, such as the stacking method and the permanent scatterers (PS) method [9]. By stacking and averaging multiple SAR interferograms, the stack method reduces the impact of the atmospheric delay effect on SAR interferometry imaging. In general, however, the stacking method reduces the time resolution of SAR measurements and may eliminate valuable geophysical signals [10]. Conversely, the permanent scatterer method estimates the atmospheric delay phase in the absence of atmospheric water vapour data by analysing the target points (PS points) where the scattering characteristics remain stable in the InSAR interferogram over a certain time range according to the spatial correlation of the atmospheric delay, thereby reducing the effect of atmospheric delay on the final ground deformation results [13]. When estimating the atmospheric phase component, however, this method must still assume that the deformation is linear or subject to periodic changes [14]. Consequently, this technique is generally ineffective when studying coseismic or post-seismic deformation. The second type uses external data to mitigate atmospheric effects [8]. Typical methods include the ground meteorological information modelling method and GPS observation correction [15]. In addition, the surface meteorological information modelling method is used to estimate the tropospheric delay based on surface meteorological observation data (humidity, temperature, pressure, etc.) to remove atmospheric effects [16]. However, this method is plagued by the sparse distribution of meteorological stations, long observation period, and low accuracy of tropospheric delay estimation, thereby resulting in low applicability in atmospheric phase delay correction [16]. The GPS observation correction method employs GPS observations to estimate zenith delay and interpolation to obtain an InSAR tropospheric delay correction map [12]. Nevertheless, this method requires dense GPS networks in most of the world. Consequently, in the case of limited GPS data, it is necessary to combine GPS and InSAR observations to obtain a high-precision and high-spatial-resolution coseismic deformation field [17,18,19,20]. Considering the complex terrain structure in the Jiashi area and the limited number of images and external data obtained covering the area, the current study, therefore, used the iterative tropospheric decomposition method to correct the atmospheric delay phase.

In summary, based on the structural and topographic characteristics of the Jiashi area, the influence of the tropospheric delay on the results of seismic coseismic deformation, and the ascending and descending Sentinel-1A SAR data, this study carried out the tropospheric delay correction, coseismic deformation field extraction, and coseismic slip distribution parameters inversion of the *Ms* 6.4 earthquake that occurred in Jiashi, Western China on 19 January 2020. The primary contributions of our work are as follows: (1) An improved inverse distance weighted interpolation tropospheric decomposition optimisation model was proposed to improve the precision of turbulence component delay estimation and realise tropospheric atmospheric delay phase correction. (2) The coseismic deformation field and deformation characteristics of the Jiashi earthquake were successfully obtained by using ascending and descending SAR data. (3) The fault slip distribution parameters of the Jiashi earthquake were inverted using coseismic deformation field data. Correspondingly, the results of this study can support and serve as a reference for the extraction of the coseismic deformation field and inversion of fault slip distribution parameters.

## 2. Materials and Methods

### 2.1. Tectonic Background

On 19 January 2020, at 21:27:00 UTC, a moderate earthquake measuring *Ms* 6.4 occurred in Jiashi County, China. The long axis of the isoseismal line caused by the earthquake was east–west, and the focal mechanism of the earthquake comprised thrust. Following the earthquake, various institutions reported focal mechanism solution outcomes (Table 1). The earthquake occurred at the collision front of two tectonic units in the Southwestern Tianshan Mountains and the Tarim Basin, and the height difference between its northern and southern regions was significant. The Tianshan Mountains are an inland orogenic belt and are the largest regenerative orogenic belt in Eurasia [21]. Their current crustal shortening rate is roughly half the average collision convergence rate of the Indian and Eurasia plates (~42 mm/year) [22]. Compressive tectonic deformation since the Late Cenozoic has resulted in the formation of a large number of E-W trending and complex reverse fault fold belts in the interior and front of the Tianshan orogenic belt [23]. In addition, the Tarim Basin is a rigid block with less seismic activity [24]. Even though the block moves northward in both the Southwestern Tianshan Mountains and the Tarim Basin, the rate of deformation in the Southern Tianshan Mountains is significantly higher than that of the Tarim Basin [25]. Moreover, there are differences in the amplitude of the neotectonic movement [25]. Thus, the Southern Tianshan Mountains continue to rise slightly, whereas the Tarim Basin correspondingly subsides. This movement will almost certainly increase the elevation difference in this region. In addition, it will result in a significant stress difference at the structural boundary, thereby forming a region with high seismicity.

The epicentre of the Jiashi earthquake was located near the Keping fault on the southern edge of the Kepingtag thrust nappe structure, in front of which the earthquake occurred (Figure 1a). The field geological survey and structural interpretation revealed that the Kepingtag thrust nappe structure is composed of multiple groups of parallel monocline and anticline mountains and their front thrust faults and that a southward protruding arcuate nappe structure has been formed on the plane (Figure 1b). From 1997 to 2003, there were three significant strong earthquake events in the Jiashi region, namely the Jiashi strong earthquake swarm of seven earthquakes in 1997, the *Ms* 6.1 and *Ms* 6.4 earthquake sequences on 2 August 1998, and the *Ms* 6.8 earthquake sequences on 24 February 2003 [26]. These three strong earthquake swarm events were concentrated in the Kepingtag fold belt and the Ozgertau fold belt, and their spatial distribution characteristics indicate that the seismic activity in the western region of the Puchang fault belt is significantly stronger than that in the eastern region, exhibiting different temporal and spatial characteristics and focal mechanism. Jiashi experienced another *Ms* 6.4 earthquake on 19 January 2020. According to the China Earthquake Networks Centre (CENC), the epicentre was located at 39.83° N and 77.21° E with a focal depth of approximately 16 km. In addition, the focal mechanism solutions reported by the United States Geological Survey (USGS) and the CENC suggest that this earthquake was most likely a thrust event in the fold belt on the southern margin of the Kepingtag nappe structural belt (Table 1). Therefore, it is essential to comprehend the seismogenic structure, mechanism, and seismic hazard of the Kepingtag thrust fold structural belt.

### 2.2. InSAR Data

The European Space Agency operates the earth observation satellite Sentinel-1 as part of the Global Monitoring for Environment and Security (GMES) program. Sentinel-1 is comprised of Sentinel-1A and Sentinel-1B. In this study, C-band (radar wavelength of 5.6 cm) Sentinel-1A images were used for both the ascending and descending tracks to measure the displacement of the Jiashi earthquake. The data were collected utilising a very advanced scanning mode known as Terrain Observation with Progressive Scans (TOPS) in the IW mode, which can effectively reduce the scalloping effect and enhance the interference performance. The detailed parameters of each interferometric pair are listed in Table 2. Due to the lack of vegetation in the epicentre region and the small temporal and spatial baseline, the coherence is high.

### 2.3. Study Methods

#### 2.3.1. Improved Inverse Distance Weighted Interpolation Tropospheric Decomposition Method

The traditional iterative tropospheric decomposition method estimates the turbulence component using the inverse distance weighted (IDW) method [27]. This method uses the reciprocal of the horizontal distance between the sampling point and the interpolation point as the weight, disregards the characteristics of the turbulence component as they vary with elevation, and is therefore unsuitable for regions with significant elevation changes. In order to accurately estimate the turbulence component in tropospheric delay, an improved inverse distance weighted interpolation tropospheric decomposition method is suggested. Typically, the turbulence component consists of medium- and long-wave signals, which can be calculated using enhanced inverse distance weighted interpolation. Assuming *n* GPS stations exist in the study area, the enhanced IDW model can be expressed as follows:(1)$$T_{u}=\overset{n}{\underset{i=1}{\sum }}w_{ui}T\left(x_{i}\right)$$
(2)$$w_{ui}=d_{ui}/\overset{n}{\underset{i=1}{\sum }}d_{ui}^{-2}$$
(3)$$d_{ui}=\sqrt{{\left(ad_{ui}^{s}\right)}^{2}+{\left(\left(1-a\right)d_{ui}^{h}\right)}^{2}}/a$$
where $w_{ui}$ represents the interpolation coefficient; u and i represent the observation station and the reference station, respectively; $d_{ui}$ indicates the distance between the reference station and the observation station; $d_{ui}^{s}$ and $d_{ui}^{h}$ represent the horizontal distance and the elevation difference between the observation station and the reference station, respectively; a is the influence factor, representing the difference in the influence degree of the horizontal and vertical distance between the observation station and the reference station on the interpolation results. The optimal value of the influence factor a can be determined using numerous experiments. Accordingly, Equations (1)–(3) constitute the function relation of the improved inverse distance weighted interpolation method.

The following are the processing steps for the enhanced inverse distance weighted interpolation tropospheric decomposition:

(a) Assuming that the turbulence component in the iterative tropospheric decomposition model is 0, the vertical component index coefficient and vertical component delay at sea level in the study area can be initially estimated based on the total zenith delay observed by all GPS stations to obtain the vertical component index coefficient;

(b) The residual component consists of the unmodelled component and the turbulence component, which can be obtained by subtracting the vertical component from the site-by-site iterative tropospheric decomposition model;

(c) Using the enhanced IDW model, the turbulence component can be determined from the residual component:(4)$$\left[\begin{array}{c} T_{1}\\ T_{2}\\ \cdots \\ T_{n} \end{array}\right]=\left[\begin{array}{cccc} 0 & w_{12} & \cdots & w_{1n}\\ w_{21} & 0 & \cdots & w_{2n}\\ \cdots & \cdots & 0 & \cdots \\ w_{n1} & \cdots & w_{n,n-1} & 0 \end{array}\right]\left[\begin{array}{c} \varepsilon_{1}\\ \varepsilon_{2}\\ \cdots \\ \varepsilon_{n} \end{array}\right]$$

(d) By subtracting the updated turbulence component from the iterative tropospheric decomposition model of each station, the index coefficient and vertical component delay at sea level in the study area were recalculated;

(e) Steps (b)–(d) were repeated until the exponential coefficient and the vertical component at the sea level of the study area delayed the convergence. The output results include the turbulence component, the vertical component, and the residual component at each observation station. The iterative tropospheric decomposition model value at the location of interest is then obtained by improved inverse distance weighted interpolation.

#### 2.3.2. Inversion of Fault Geometry

The Monte Carlo algorithm is a prevalent method for solving inversion problems [28]. It is an estimation method that utilises the linear iterative method to solve nonlinear problems and maximises the posterior probability density function of the estimated parameters, considering the prior information of the model parameters and continuously perturbing the model parameters in conjunction with the Metropolis–Hasting (M–H) algorithm [29,30,31]. When compared to other nonlinear optimisation algorithms, such as the simulated annealing algorithm, particle swarm optimisation algorithm, and genetic algorithm, the Monte Carlo algorithm has the advantages of rapid iteration and straightforward calculation. The current study utilises the Monte Carlo algorithm to invert fault geometric parameters in order to obtain precise fault dislocation slip data. The expression of the Monte Carlo algorithm used to invert geometric parameters of the fault is as follows:(5)$$p\left(m|d\right)=f\left(m\right)+e$$
where $p\left(m|d\right)$ is the posterior probability density function of the geometric parameters to be inverted, *m* is the initial constraint of the fault’s geometric parameters, d is the obtained seismic coseismic deformation data, f is the forward operator, and e is the noise component of the observed deformation data. The joint likelihood function $p\left(d|m\right)$ and a priori information $p\left(m\right)$ in the Bayesian frame of Equation (1) can be written as follows:(6)$$p\left(m|d\right)=p\left(m\right)\times p\left(d|m\right)$$

The likelihood function needs to be established using the M-H global optimisation algorithm. Assuming that it obeys a Gaussian distribution, the mean and standard deviation are $\mu_{m}$ and $\sigma_{m}^{2}$, respectively. Subsequently,
(7)$$p\left(m\right)=\frac{1}{{\left(2\pi \sigma_{m}^{2}\right)}^{\frac{N}{2}}}\exp\left[-\sum \frac{{\left(m-u_{m}\right)}^{2}}{2\sigma_{m}^{2}}\right]$$

In general, the seismic background noise conforms to the normal distribution with a mean value of 0 and a variance of $\sigma_{n}^{2}$ for the likelihood function. By constraining the fault’s geometric parameters based on the likelihood function, the lateral continuity and stability of the inversion can be enhanced. The likelihood function can be, therefore, expressed as follows:(8)$$p\left(d|m\right)=\frac{1}{{\left(2\pi \sigma_{n}^{2}\right)}^{\frac{N}{2}}}\exp\left[-\sum \frac{{\left(d-f\left(m\right)\right)}^{2}}{2\sigma_{n}^{2}}\right]$$
where *N* is the number of sampling points. Subsequently, Equation (2) can be rewritten as
(9)$$p\left(m|d\right)=\lambda \exp\left[-\sum \frac{{\left(d-f\left(m\right)\right)}^{2}}{2\sigma_{n}^{2}}-\varepsilon \sum \frac{{\left(m-\mu_{m}\right)}^{2}}{2\sigma_{m}^{2}}\right]$$
where ε is an adjustable factor, and λ is a constant.

To obtain the posterior probability density function distribution $p\left(m|d\right)$ of the estimated fault parameters, the M-H algorithm is used to generate a Markov chain. In general, the distribution function obeys a Gaussian distribution:(10)$$q\left(m_{k},m^{*}\right)\sim N\left(0,\sigma_{m}^{2}\right)$$
where $q\left(m_{k},m^{*}\right)$ represents the potential transfer of the initial model value $m_{k}$ when the sampling point is k. The transition probability can be expressed as follows:(11)$$a\left(m_{k},m^{*}\right)=min\left[1,\frac{\pi \left(m^{*}\right)q\left(m^{*},m_{k}\right)}{\pi \left(m_{k}\right)q\left(m_{k},m^{*}\right)}\right]$$

The posterior probability density distribution must satisfy the stationary distribution $p\left(m\right)\times p\left(d|m\right)$ for the Markov chain to converge to the posterior probability distribution of unknown parameter *m*. The probability of a transition is then calculated as follows:(12)$$a\left(m_{k},m^{*}\right)=\exp\left[\alpha^{\prime }\left(m_{k},m^{*}\right)\right]$$
where, $J\left(m_{k}\right)$ and $\alpha^{\prime }\left(m_{k},m^{*}\right)$ are defined as in Equations (13) and (14), respectively.
(13)$$J\left(m_{k}\right)=-\sum \frac{{\left[d-f\left(m_{k}\right)\right]}^{2}}{2\sigma_{n}^{2}}-\varepsilon \sum \frac{\left(m_{k}-\mu_{m_{k}}\right)}{2\sigma_{m_{k}}^{2}}$$
(14)$$\alpha^{\prime }\left(m_{k},m^{*}\right)=\min\left\{0,J\left(m^{*}\right)+\mathrm{lg}\left[q\left(m^{*},m_{k}\right)\right]-J\left(m_{k}\right)-\mathrm{lg}\left[q\left(m_{k},m^{*}\right)\right]\right\}$$

Finally, the model parameters were continuously disturbed under the M-H sampling algorithm. Accordingly, multiple random simulations were performed on the initial model parameters of the fault’s geometry, and the posterior mean was used as the optimal solution for the model parameters.

#### 2.3.3. Inversion of Fault Slip Distribution

To comprehend the fine slip distribution on the fault plane, the fault geometry parameters are obtained via nonlinear inversion. Accordingly, we set the fault plane to be 50 km long along the strike and 30 km wide on the basis of the distribution range of the coseismic deformation field, and it was subdivided into 375 sub-faults, each measuring 2 × 2 km^2^ [32]. As a result of determining the geometric parameters of the fault, the slip distribution on the fault plane was linearly correlated with the deformation data, allowing it to be solved by linear inversion. In addition, it is necessary to impose certain smoothing constraints on the observation equation in order to avoid matrix rank deficiency and the resulting concussion in the solution process. The equation can, thus, be expressed as:(15)$$\left[\begin{array}{c} d\\ 0 \end{array}\right]=\left[\begin{array}{c} G\\ k^{2}D \end{array}\right]m+\left[\begin{array}{c} \varepsilon \\ 0 \end{array}\right]$$
where *d* is the observed LOS displacement from InSAR, *G* is the Green function linking the predicted displacement to the unit model slip, m is the slip amount of sub-fault, including the strike-slip and dip-slip components, $k^{2}$ is the smoothing factor, *D* is the finite difference approximation of the Laplacian operator, which was used to avoid a slip distribution characterised by nonphysical oscillations, and ε is the observation error.

## 3. Results

### 3.1. Coseismic Deformation Field Result

Using two-track differential interferometry and the GAMMA software, the coseismic deformation fields of the ascending and descending Sentinel-1A tracks were obtained. To reduce phase noise, interferograms were downsampled to ten looks in the azimuth and two looks in the range and filtered using an adaptive filter function. The accuracy of the azimuth registration (better than 0.001 pixel) was acquired to prevent phase jumps between successive bursts, and the effects of topography were removed from the interferograms using a Shuttle Radar Topography Mission (SRTM) digital elevation model (DEM) with a 30 m resolution. In addition, the minimum-cost flow (MCF) method was utilised for phase unwrapping. Figure 2 represents the differential interferograms of the 2020 *Ms* 6.4 Jiashi earthquake based on ascending and descending images from Sentinel-1A. According to Figure 2, we can deduce the following information. First, all of the interference fringes are smooth, and there are no significant areas of decorrelation, indicating that the overall coherence of this region is satisfactory. All interference fringes have an elliptical distribution along the north–south axis and are asymmetric in the ascending and descending tracks. In addition, the colour order of the southern fringes is the opposite of that of the northern fringes, and the number of deformation fringes in the south is greater than that in the north, indicating that the deformation trend in the south is the opposite of that in the north and that the magnitude of deformation in the south is greater than that in the north. Even though the primary orbital error and the atmospheric error have been filtered out of the differential interferogram, there are still some atmospheric residual phases in a region far from the seismic deformation field.

The Jiashi earthquake occurred in the Kepingtag thrust nappe structural belt between the Southern Tianshan Mountains and the Tarim Basin; this region is characterised by strong topographic fluctuations; the influence of the tropospheric delay phase is significant; therefore, residual atmospheric phases remain in the interferogram acquired with the DInSAR. We used the improved IDW interpolation tropospheric decomposition method to remove atmospheric effects in order to reduce the influence of atmospheric effects even further. Figure 3 depicts the seismic deformation field prior to and subsequent to its removal.

Clearly, the deformation field caused by the earthquake was predominantly distributed in the southernmost region of the Kepingtag thrust nappe structural belt, and it was concentrated in the Kepingtag fold belt and the Ozgertau fold belt. From the perspective of space, the long-axis direction of the deformation field is nearly EW, indicating that the strike of the seismogenic fault is nearly EW. There are two obvious deformation areas in the LOS deformation fields: the deformation in the Ozgertau fold belt was subsidence, with a maximum value of approximately 0.04 m, while an uplift LOS deformation with a maximum value of 0.06 m is located in the Kepingntag fold belt. The displacement between the two deformation areas is continuously distributed, and there is no decoherence area caused by surface rupture, indicating that the rupture of the seismic fault did not reach the surface. The signs of the LOS deformation observed in the descending and ascending deformation fields are identical, indicating that the seismogenic fault deforms primarily vertically. Moreover, the uplift in the south (south wall) is significantly greater than the subsidence in the north (north wall), indicating that the seismogenic fault dips northward, which is consistent with the focal mechanism solutions reported by USGS, GCMT, and other institutions [3]. Since the earthquake occurred in the low-angle thrust nappe structural belt, combined with the focal mechanism solution and the regional structural characteristics, we hypothesised that the seismogenic fault of the Jiashi earthquake might be the Kepingtage fault on the Kepingtage thrust nappe tectonic belt.

### 3.2. Inversion of Fault Geometry and Slip Distribution

In this study, the optimal geometric parameters of the fault were determined using the Monte Carlo method and ascending and descending deformation data as constraints. In addition, the uniform fault sliding model and the joint probability density distribution of the Monte Carlo Markov chain were obtained. Finally, the optimal geometric model of the fault was obtained. In summary, the uniform slip results provide a single fault geometry of 50 km in length and 31 km in width. The strike, dip, and rake angles are 270°, 22°, and 90°, respectively, which are comparable to the USGS-reported focal mechanism solution. In addition, the strike is consistent with the long axis of the interferograms, and the dip angle is consistent with the available geological data.

We inverted the slip distribution of the Jiashi earthquake using the Steepest Decent Method (SDM) and finally obtained the slip distribution of the Jiashi earthquake (Figure 4). Significant slip area was primarily concentrated in a fault plane measuring 30 km in length and 20 km in width. The coseismic slip was concentrated at depths between 10 and 20 km, with the maximum slip occurring at approximately 15 km. Peak slip of 0.34 m occurred at 39.90° N, 77.30° E, 5.3 km below the Earth’s surface, and the slip angle was 92.71°. The average slip of the seismogenic fault obtained by inversion is 0.04 m. The results of the inversion indicate that the fault has a strike of 274.87°, a dip of 20°, and an average rake of 90.59° (Table 3). The determined slip model, assuming a shear modulus of 30 GPa, has a total moment of 6.93 × 10^18^ N·m, which corresponds to a moment magnitude of 6.06, which is consistent with the seismic moments reported by the USGS and GCMT. The fault slip model indicates that the Jiashi earthquake was a typical thrust fracture event with a small component of strike slip. The source parameters of the fault geometry inversion and the slip distribution inversion are extremely similar, and the results of the two inversions can be independently validated.

The Jiashi earthquake fault activity did not cause a surface fracture, and the thrust strike-slip Keping fault is located near the source area (Figure 3). In addition, the aftershock relocation results show that the aftershocks are mainly distributed along the fault strike, most of them are concentrated at a depth of 5–20 km underground, and the focal depth gradually deepens from north to south, with an obvious N-dip trend, which is also consistent with the inversion results. Moreover, the inversion results in this paper are in quite good agreement with the research results of Wen and Lee [4,6]. The above analysis and conclusions also confirm the reliability of the results of this paper.

Figure 5 depicts the observed, modelled, and residual (the observation value minus prediction value) deformations derived from the descending and ascending Sentinel-1A data. In terms of distribution shape, deformation magnitude, and movement, the inverted deformation fields of the ascending and descending tracks are highly consistent with the observed deformation fields, indicating the rationality and dependability of the earthquake fault slip model. In addition, there is a small residual (0.05 m) in the piedmont of the southwestern edge of the Kepingtag fold, which may be due to the DEM error and unwrapping error resulting from the large topographic fluctuations between the Kepingtag fold belt and the Tarim Basin. In addition, multiple studies have demonstrated that the zenith tropospheric delay changes in the form of a negative exponent with increasing elevation; thus, the correction values of the zenith tropospheric delay in the northern and southern portions of the deformation area are quite different, which may also result in a large residual value in the south. The maximum uplift and subsidence values of ascending InSAR observations were determined to be 0.06 m and 0.04 m, respectively, whereas the simulated values indicate that the maximum uplift was 0.06 m and the maximum subsidence was 0.03 m. The maximum uplift and subsidence values of descending InSAR observations were determined to be 0.05 m and 0.04 m, respectively, whereas the simulated values indicate that the maximum uplift was 0.06 m and the maximum subsidence was 0.04 m. In the deformation region, the simulated residuals of ascending and descending orbits are less than 3 cm. The correlation coefficient between the observations and predictions is 99.6%, indicating that the fault slip model developed in this study has a high degree of congruence with the observed data and that the resulting slip distribution is more reliable.

## 4. Discussion

The observed and modelled deformation were found to be generally consistent, indicating that there is no break exposed at the surface. In addition, it suggests that the fault closest to the focal area is the Keping fault, which is a thrust slip fault. The results of the relocated Jiashi earthquake sequence indicate that the aftershocks were primarily distributed along the strike of the fault and concentrated between 5 and 20 km beneath the surface. The depth of the focal region gradually increased from north to south, with a distinct northward dip. According to the focal mechanism solutions reported by various institutions for the Jiashi earthquake (Table 1), the fault’s strike (274.87°) and dip (20°) obtained via slip distribution inversion utilising the SDM program are comparable to the mechanism solution reported by the USGS (strike = 221°, dip = 20°). The rake angle (90.59°) is close to the mechanism solution reported by the GFZ (rake angle = 94°), which agrees with the findings of Wen et al. [6]. According to the above preliminary analysis and the fault slip model derived from the coseismic deformation field, we believe that the Jiashi earthquake was a fracture event dominated by a northward-dipping thrust fault with a small component of strike slip. The earthquake-causing structure was the Kepingtag thrust fault in the Kepingtag fold belt.

Furthermore, both the 1997–2003 Jiashi earthquake swarm and the earthquake occurred within the Southern Tianshan thrust nappe structural belt. The thrust nappe has the Keping fault and a concealed thrust fault zone as its tectonic background. The formation of the concealed fault zone may have resulted from the heterogeneous deformation and ongoing tectonic movement along the northern margin of the Tarim Basin. Some faults extend beneath the Tarim Basin and are concealed by Cenozoic sedimentary layers, thereby forming concealed faults. These concealed faults can generate earthquakes of moderate intensity. In the thrust nappe structural belt, a small-scale normal fault may occur; a transverse strike-slip fault may even form nearly perpendicular to the strike of the thrust fault beneath the thrust nappe structure due to differences in the mechanical properties of the rocks and the regional tectonic stress fields. These normal faults and strike-slip faults are also capable of causing moderately powerful earthquakes. Consequently, based on the analysis of the regional tectonic dynamic background and the regional tectonic distribution, we believe that the occurrence mechanism of the Jiashi *Ms* 6–7 earthquake swarm from 1997 to 2003 and the Jiashi *Ms* 6.4 earthquake should be distinct rupture events in the foreland tectonic environment of the Southern Tianshan Mountains. They were identical occurrences brought on by tectonic activity in the thrust nappe structure of the Southern Tianshan Mountains.

## 5. Conclusions

This study proposes an improved IDW interpolation tropospheric decomposition optimisation model to estimate the turbulence component in the tropospheric delay in order to reduce the effect of atmospheric residual phase on the spatial distribution of the LOS surface deformation fields with Sentinel-1 data interferometry in the Jiashi earthquake. Using ascending and descending data from Sentinel-1A, the LOS deformation fields of the earthquake were then extracted. Based on this information, the fault geometry and slip distribution of the earthquake are inverted. In particular, the results presented in the paper demonstrate that:

(1) The coseismic deformation field of the Jiashi earthquake was approximately 60 km wide in the strike direction and 50 km wide in the dip direction. In the southern and northern parts of the deformation field, there were two deformation centres with nearly elliptical distributions. The maximum northward subsidence was 0.04 m, while the maximum southward uplift was 0.06 m. The absence of a spatial decoherence zone and the continuity of the interference fringes between the subsidence area and the uplift area indicate that the seismic rupture did not reach the earth’s surface. The results of the inversion indicate that the fault has a strike of 274.87°, a dip of 20° N, an average rake of 90.59°, and a maximum slip of approximately 0.34 m at a depth of 5.3 km. The total geodetic moment for a shear modulus of 30 GPa was 6.93 × 10^18^ N·m, equivalent to *Ms* 6.06. The Jiashi earthquake was a northward-dipping thrust earthquake with a small strike-slip component, which was consistent with the geological tectonic background of the region. In addition, based on the focal mechanism and regional structural characteristics, we conclude that the seismogenic structure of the Jiashi earthquake could be the Keping fault at the leading edge of the Kepingtag nappe structural belt.

(2) The earthquake that struck Jiashi on 19 January 2020, was a typical thrust rupture event that occurred in the foreland of the South Tianshan Mountains. Between 1997 and 2003, this region experienced an intense swarm of *Ms* 6–7 earthquakes. The comprehensive analysis of the regional tectonic dynamic background and the deep and shallow regional structures led us to the preliminary conclusion that the seismogenic mechanisms of the Jiashi *Ms* 6.4 earthquake and the Jiashi *Ms* 6–7 earthquake swarm from 1997 to 2003 were fault rupture events formed by cracks at different levels in the foreland environment of the Southern Tianshan Mountains, which are jointly controlled by the thrust nappe of the Southern Tianshan Mountains.

(3) Using DInSAR deformation observation data, the geometric and kinematic parameters of the seismogenic fault of the Jiashi earthquake were inverted, and a preliminary explanation for the dynamic physical mechanism of the seismogenic fault was provided, which has significant implications for earthquake prevention and disaster reduction. In the following step, we will combine GPS and other data to conduct research on source parameters and source rupture process inversion, as well as discuss the change in stress distribution after the earthquake, to comprehensively analyse scientific problems such as co-earthquake and post-earthquake stress and strain.

## Figures and Tables

**Figure 1 sensors-23-03046-f001:**
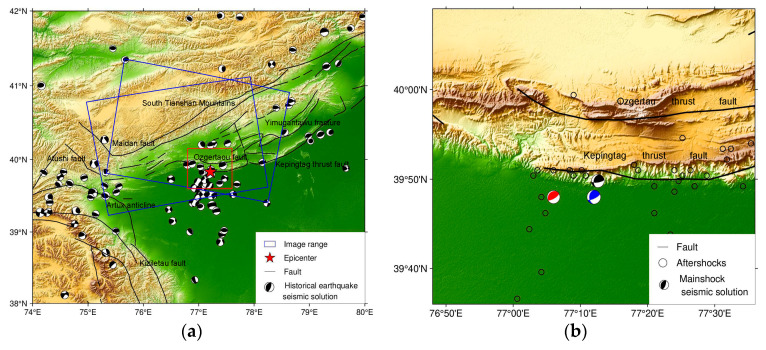
Tectonic background of the 2020 *M_s_* 6.4 Jiashi earthquake. (**a**) Regional map showing the location of the earthquake. (**b**) Tectonic setting. The blue rectangles mark the InSAR data coverage; the black beach balls denote the *M*_s_ > 3 earthquakes that occurred in the area from 1965 to 2020; the red star denotes the epicentre; the red rectangle outlines the area shown in (**b**). The red, black, and blue beach balls show the focal moment solutions reported by the USGS, Global Centroid Moment Tensor (GCMT), and CENC, respectively, which are shown in (**b**). The black dots represent the aftershocks (*M*_s_ > 3) that occurred between 19 and 26 January 2020, reported by the GCMT. The black lines depict the major strike fault, Keping fault, and Ozgertaou fault.

**Figure 2 sensors-23-03046-f002:**
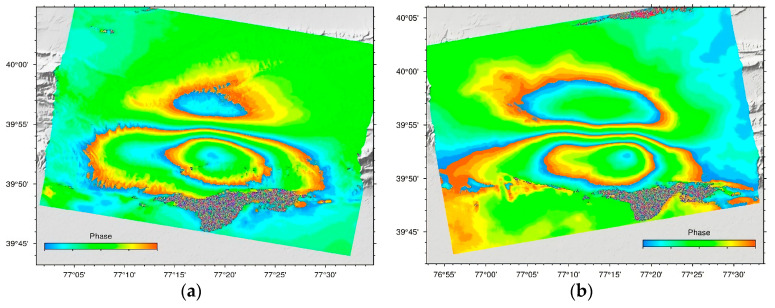
The line-of-sight (LOS) coseismic differential interferograms of the 2020 Jiashi *Ms* 6.4 earthquake based on the ascending and descending images from Sentinel-1A. (**a**,**b**) show the descending and ascending differential interferograms, respectively. Each cycle of colour (from blue to red) represents a half radar wavelength (2.8 cm) along the LOS direction.

**Figure 3 sensors-23-03046-f003:**
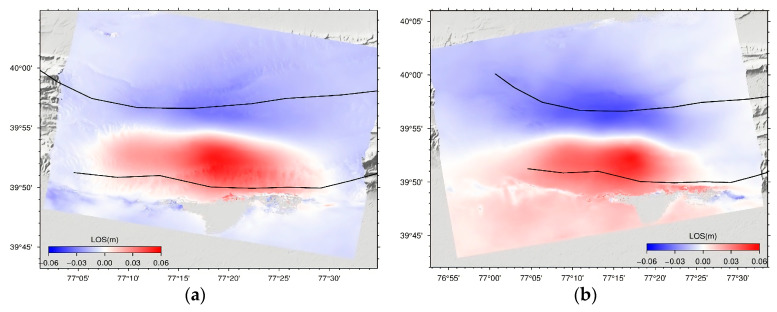
Deformation fields of the Jiashi *Ms* 6.4 earthquake acquired using Sentinel-1A data for the descending and ascending tracks. (**a**,**b**) show the descending and ascending LOS coseismic deformation maps without tropospheric delay correction, respectively. (**c***,***d**) show the descending and ascending LOS coseismic deformation maps after atmospheric correction, respectively. The black lines denote the main faults. The positive values indicate that the earth’s surface moves toward the LOS direction.

**Figure 4 sensors-23-03046-f004:**
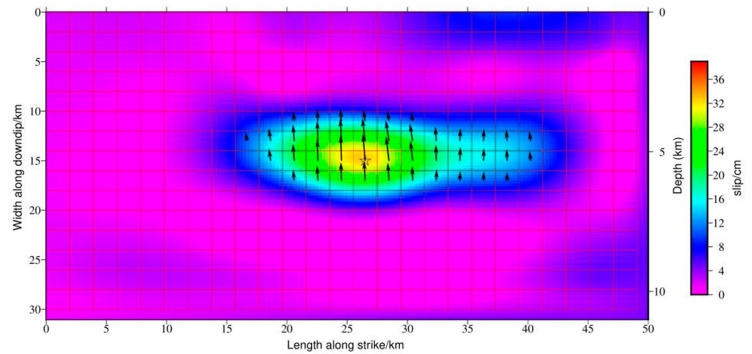
Slip distribution of the Jiashi earthquake inverted from InSAR data. The yellow star represents the Retrieved Epicenter Position. The slip direction of the coseismic deformation field is indicated by black arrows. The size of the arrow represents the amount of sliding.

**Figure 5 sensors-23-03046-f005:**
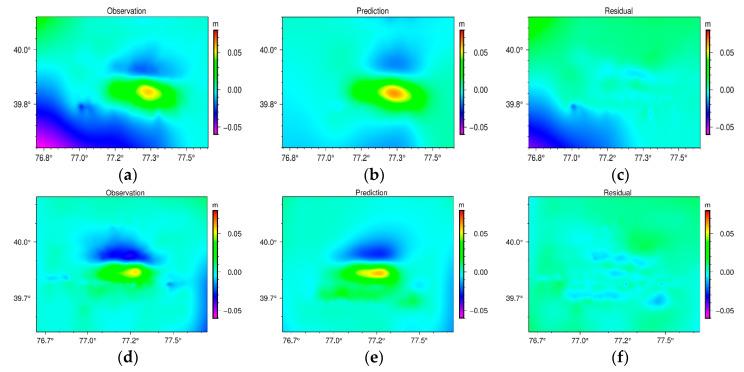
Coseismic displacement (negative values indicate range displacement away from the satellite) and model for the slip distribution inversion of the 2020 Jiashi earthquake. (**a**–**c**) are the measurement, model, and residual derived from the descending track. (**d**–**f**) are the measurement, model, and residual derived from the ascending track.

**Table 1 sensors-23-03046-t001:** Focal mechanisms and fault parameters reported by different institutions.

Lat (°)	Lon (°)	Depth (km)	Mag	Nodal Plan I			Nodal Plan II			Agency
				Strike	Dip	Rake	Strike	Dip	Rake	
39.8	77.2	12	6.0	196	37	30	81	72	123	GCMT
39.8	77.1	20	6.0	221	20	72	60	71	96	USGS
39.8	77.2	16	6.4	182	35	32	65	72	121	CENC
39.8	77.1	16	6.1	56	75	94	222	16	77	GFZ

**Table 2 sensors-23-03046-t002:** Coseismic interferometry pairs.

Flight Direction	Track	Master	Secondary	Time Interval	Perpendicular Baseline
Descending	T034	10 January 2020	22 January 2020	12	57
Ascending	T129	16 January 2020	28 January 2020	12	11

**Table 3 sensors-23-03046-t003:** Focal mechanism solutions of the Jiashi earthquake.

Source	Lon	Lat	Length	Depth	Width	Strike	Dip	Rake ^1^	Slip ^2^	Mw
	(°E)	(°N)	(Km)	(Km)	(Km)	(°)	(°)	(°)	(m)	
InSAR	77.30	39.90	50	10	31	274.87	20	90.59	0.34	6.06
USGS	77.11	39.835	-	20	-	221	20	72	-	6.03
GCMT	77.18	39.78	-	12	-	196	37	30	-	6.0
CENC	77.21	39.83	-	16	-	182	35	32	-	6.0
GFZ	77.10	39.80	-	16	-	222	16	77	-	6.1
Li et al. [17]	-	-	58	10	30	270	15	85	0.34	6.0
Yu et al. [5]	77.30	39.89	50	4.97	20	275	17	84.96	0.29	6.09

^1^ Mean rake direction determined by each fault patch; ^2^ Maximum slip in the fault plane.

## Data Availability

The authors confirm that the data supporting the findings of this study are available within the article.

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
