# Peer review of "Coseismic Deformation Field and Fault Slip Distribution Inversion of the 2020 Jiashi Ms 6.4 Earthquake: Considering the Atmospheric Effect with Sentinel-1 Data Interferometry"

_sensors, 2023, doi:10.3390/s23063046_

Round 1

Reviewer 1 Report

In this study, an improved inverse distance weighted interpolation tropospheric decomposition optimization model was proposed to improve the precision of turbulence component delay estimation and realize tropospheric atmospheric delay phase correction. Based on that, the coseismic deformation field and deformation characteristics of the Jiashi earthquake were successfully obtained by using ascending and descending SAR data. The results of this study can support and serve as a reference for the extraction of coseismic deformation field and inversion of fault slip distribution parameters. However, the innovations or improvements of the method applied in this study different from previous studies need to be clearly represented. Besides, the analyses of the seismogenic structure and behavior of the earthquake were still flawed. I encourage to further review the manuscript after minor revision.

The suggestions are listed below.

1. In the study methods, how many GPS stations are used for calculating the tropospheric delay? The GPS data used in this study should also be illustrated in the Materials part.

2. The Monte Carlo algorithm is an existing method that can be used for fault geometry inversion rather than a new method proposed by the authors. Therefore, more related references should be added in the “2.3.2 Inversion of Fault Geometry” part.

3. In Figure 4, the meaning of the yellow star and black arrows need to be clearly represented.

4. In the Discussion part, pictures of the regional tectonic dynamic background and its relationship with the deformation should be added to support more intuitive analysis of the seismogenic structure and behavior of the earthquake. 

Reviewer 2 Report

The objective of this paper is analyzing the coseismic deformation field and inversion of fault slip distribution parameters, related to the 2020 Jiashi Ms 6.4 Earthquake.

This is an interesting and well-structured paper. All necessary sections (Introduction, Materials and Methods, Results, Discussion, Conclusions) have been considered. Moreover, the “Materials and Methods” and “Results” sections are divided into sub-sections, providing additional details. Furthermore, all Figures and Tables are consistent with the analysis provided in the manuscript. Regarding the mathematical part, described in the “Study methods” sub-section, it is valid (the necessary analytical equations explanation is also provided). However, I suggest some changes, which will overall improve the paper. In particular:

Lines 11-33: Although the abstract has been properly structured, many details have been included. The abstract should be more concise and include the most significant information. All additional details should be removed and placed in the manuscript’s main body. Please, modify.

Line 84: The manuscript, before this line, describes the InSAR and GPS observations contribution in the coseismic deformation. After this line a brief paragraph should be added, in which the combination of InSAR and GPS observations will be analyzed. Indicative papers, in which the corresponding information can be obtained, are the following: 1. He, Z., Chen, T., Wang, M., Li, Y., 2020. Multi-Segment Rupture Model of the 2016 Kumamoto Earthquake Revealed by InSAR and GPS Data. Remote Sens (Basel) 12, 3721. https://doi.org/10.3390/rs12223721, 2. Kumar Maurya, V., Dwivedi, R., Ranjan Martha, T., 2022. Site scale landslide deformation and strain analysis using MT-InSAR and GNSS approach – A case study. Advances in Space Research 70, 3932–3947. https://doi.org/10.1016/j.asr.2022.05.028, 3. Lazos, I., Papanikolaou, I., Sboras, S., Foumelis, M., Pikridas, C., 2022. Geodetic Upper Crust Deformation Based on Primary GNSS and INSAR Data in the Strymon Basin, Northern Greece - Correlation with Active Faults. Applied Sciences 12, 9391. https://doi.org/10.3390/app12189391, 4. Yalvac, S., 2020. Validating InSAR-SBAS results by means of different GNSS analysis techniques in medium- and high-grade deformation areas. Environ Monit Assess 192, 120. https://doi.org/10.1007/s10661-019-8009-8. Please, apply.

Line 148: Before Figure 1, I suggest adding a geological map, in which the major lithological formations will be included, as well as the major tectonic structures of the surrounding area. Please, apply.

Lines 272-274: Please, justify why 50 km length and 30 km width were preferred for the fault plane.

Line 309: Please, provide Figure 2 in a higher resolution.

Lines 322-323: Similarly, provide Figure 3 in a higher resolution, please.

Line 376: Please, provide a more detailed description in the Figure 4 caption.
